# MixRL: Data Mixing Augmentation for Regression using Reinforcement Learning

## Abstract

Data augmentation is becoming essential for improving regression accuracy in critical applications including manufacturing and finance. Existing techniques for data augmentation largely focus on classification tasks and do not readily apply to regression tasks. In particular, the recent Mixup techniques for classification rely on the key assumption that linearity holds among training examples, which is reasonable if the label space is discrete, but has limitations when the label space is continuous as in regression. We show that mixing examples that either have a large data or label distance may have an increasingly-negative effect on model performance. Hence, we use the stricter assumption that linearity only holds within certain data or label distances for regression where the degree may vary by each example. We then propose MixRL, a data augmentation meta learning framework for regression that learns for each example how many nearest neighbors it should be mixed with for the best model performance using a small validation set. MixRL achieves these objectives using Monte Carlo policy gradient reinforcement learning. Our experiments conducted both on synthetic and real datasets show that MixRL significantly outperforms state-of-the-art data augmentation baselines. MixRL can also be integrated with other classification Mixup techniques for better results.

## 1 Introduction

As machine learning (ML) becomes widely used in critical applications including manufacturing and finance, data augmentation for regression becomes essential as it provides an opportunity to improve model performance without additional data collection. In comparison to classification tasks like object detection in images, the goal of regression is to predict one or more real numbers.

To emphasize the importance of data augmentation in regression, we provide a case study of semiconductor manufacturing. Here a common quality check is to measure the layer thicknesses of a 3-dimensional semiconductor and see if they are even. However, directly measuring each thickness results in destroying the semiconductor itself, so a recently-common approach is to take an indirect measurement by applying light waves on the semiconductor, measuring the spectrum of wavelengths that bounce back from all the layers, and use ML to predict the layer thicknesses from the spectrum data (see Fig. 4 in Sec. 4 for an illustration). With enough spectrum data and thickness information, ML models can be trained to accurately predict thicknesses from a spectrum. The main challenge is that there is not enough training data, and the only cost-effective solution is to augment small amounts of data that exist. Even a small improvement in model performance from the data augmentation has significant impact in this industry. In general, any regression task that predicts real values like emissions, stock prices, or even someone's salary can also benefit from data augmentation.

Most data augmentation techniques are designed for image classification. In particular, Mixup (Zhang et al., 2018; Berthelot et al., 2019; Yun et al., 2019) is a popular data augmentation technique that is widely used for classification tasks, but is seldom used for regression because it assumes a distinct label space. The idea of Mixup is to mix pairs of examples using the key assumption that taking a linear interpolation between examples can be used to estimate the label of any examples in between. Mixup is known to effectively regularize the model. More recently, Manifold Mixup (Verma et al., 2019) has been proposed to improve the hidden representation and decision boundaries where two examples are mixed in multiple hidden layers of neural networks.

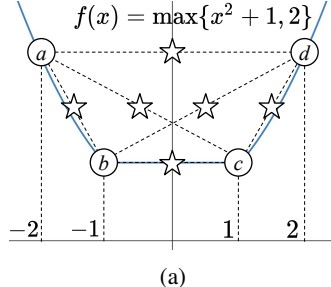 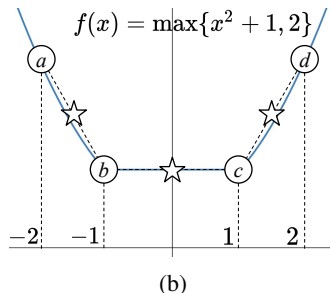

(a)                                             (b)

Figure 1: A comparison between the Mixup (Zhang et al., 2018) and MixRL on a small regression dataset where each example (circle) is associated with a single-dimensional feature (x-axis). (a) Mixup takes a linear interpolation of all possible example pairs. Unfortunately, the mixed examples (stars) do not properly reflect the function $f$ (blue plot) where some labels are arbitrarily incorrect. (b) MixRL learns for each example how many nearest neighbors it should be mixed with. As a result, each example is only mixed with its 1- or 2-nearest neighbors to better reflect $f$.

However, the Mixup techniques are not readily applicable to a regression setting because the key linearity assumption does not necessarily hold. Since the label space is continuous, taking a linear interpolation of examples that are very different either data-wise or label-wise may result in arbitrarily-incorrect labels as shown in Fig. 1a. As a result, the linearity assumption only holds to a certain extent, and the degree may vary for each example. Moreover, other data augmentation techniques for classification including image processing (e.g., flipping or rotating) and generative models (e.g., GAN (Goodfellow et al., 2014) and VAE (Kingma & Welling, 2014)) are even less applicable to a regression setting (see Sec. 5).

We propose MixRL, a data mixing augmentation framework that is the first to tailor Mixup for regression tasks using reinforcement learning. MixRL uses a stricter linearity assumption where it only holds within a certain data or label distance. These distance limits may vary by example, and we formulate the problem of learning for each example how many nearest neighbors it should be mixed with. MixRL employs a meta learning framework that estimates how valuable mixing an example is for reducing the model loss on a small validation set using Monte Carlo policy gradient reinforcement learning. MixRL's framework is inspired by the recent Data Valuation using Reinforcement Learning (DVRL) framework (Yoon et al., 2020), which solves the different problem of measuring how individual examples contribute to model performance without any mixing involved. Fig. 1b shows how limiting the nearest neighbors to mix is better than mixing with all neighbors as in classification. To see if the augmentation is useful, we train simple models on the original four examples (i.e., no augmentation), the augmented data in Fig. 1a, and the augmented data in Fig. 1b. Evaluating the models on 20 random test examples results in Root Mean Square Error (RMSE; see Sec. 4) values of 0.2967, 0.5615, and 0.1834, respectively, where a lower RMSE is better. We thus conclude that carefully mixing examples is important for improving regression performance.

Experiments conducted on real and synthetic datasets show that MixRL shows better model performances relative to baselines, especially when the linearity is limited, and the mixing must be done selectively. In addition, MixRL only requires small validation sets and scales to large training sets.

## 2 LIMITED LINEARITY IN DATA AND LABEL SPACE FOR REGRESSION

We explain why the key linearity assumption used for Mixup in classification has limitations in a regression setting. In classification, the labels are discrete where many examples may have the same label. The original version of Mixup (Zhang et al., 2018) is to take a linear interpolation between any pair of examples $x_i$ and $x_j$ with the labels $y_i$ and $y_j$ to produce the new example $\lambda x_i + (1-\lambda)x_j$ with the label $\lambda y_i + (1-\lambda)y_j$ where $\lambda \sim Beta(\alpha, \alpha)$. The linearity assumption turns out to be reasonable because the label difference between examples is only 0 or 1 and thus not that sensitive to the data difference. In contrast, the labels in regression are in a continuous space. Although there is still a many-to-one relationship where multiple examples may have the same label, the degree is much smaller than in classification. As a result, when two examples are mixed, the interpolated label can be arbitrarily different than the actual label, e.g., mixing the points $a$ and $d$ in Fig. 1a results in a

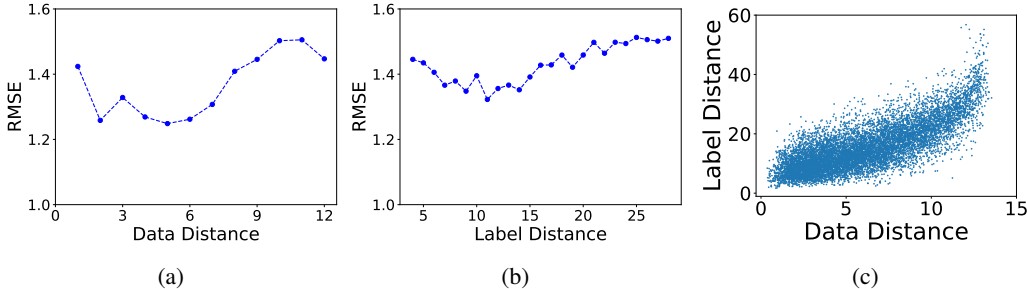

Figure 2: Using the Product dataset described in Sec. 4, (a) as the data distances between mixed examples increase, the RMSE (lower is better) of a model trained on the augmented training set initially decreases, but then gradually increases; (b) similar observations are made when increasing the label distances; and (c) the data and label Euclidean distances do not have a linear relationship.

label nowhere near the actual label. In Sec. 4.1, we also show empirical results where the label error increases for larger data or label distances. Furthermore, mixing examples with larger data or label distances tend to have increasingly-negative effects on the model trained on the augmented training set. Figs. 2a and 2b show the model accuracies using RMSE for the Product dataset (described in Sec. 4) when mixing examples with different ranges of distances. Regardless of adjusting the label or data distance, there are diminishing benefits for larger distances.

How do we limit the data and label distances to improve Mixup for regression? One approach is to only limit the data distance, which limits the label distance as well. Suppose that the regression function $f$ is continuous where $\lim_{x \to c} f(x) = f(c)$ for any x and c within the domain of $f$. We show that a short-enough data distance sufficiently reduces the label distance as well. Given $f$'s domain $D$ and $\lim_{x \to c} f(x) = L$, the following is known to hold: $\forall \epsilon, \exists \delta$ s.t. $\forall x \in D$, if $|x - c| < \delta$, then $|f(x) - L| < \epsilon$. We can use this result to prove that $\forall \epsilon, \exists \delta$ s.t. $\forall x_i, x_j \in D$, if $\alpha x_i + (1 - \alpha) x_j = c$, $0 \le \alpha \le 1$, and $|x_i - x_j| < \delta$ then the absolute difference between the mixed example's $x_j$ value and $L$ is small where $|\alpha f(x_i) + (1 - \alpha) f(x_j) - L| = |\alpha(f(x_i) - L) + (1 - \alpha)(f(x_j) - L)| \le \alpha |f(x_i) - L| + (1 - \alpha)|f(x_j) - L| < \alpha \epsilon + (1 - \alpha) \epsilon = \epsilon$. We cannot do the converse and limit the data distance by limiting label distance because there is a many-to-one mapping from data to labels, which means that two different examples may have identical labels (e.g., Fig. 1a's $a$ and $d$).

However, limiting the data distance to also limit the label distance may be too restrictive because there is not much correlation between the data and label distances in real data. Fig. 2c shows how the data distance relates to the label distance for the Product dataset. Even for small data distances, the label distance has a large range, which means that the data distance would have to be extremely limited. Hence, our solution is to limit the data and/or label distance as needed instead of just the data distance. This approach turns out to be more practical as we demonstrate in Sec. 4.4.

## 3 MIXRL

The goal of MixRL is to identify which examples to mix with which nearest neighbors. Instead of finding the actual distance limits themselves, we solve the identical problem of finding the number of nearest neighbors to mix per example for convenience. We use reinforcement learning because there is no training data on how mixing each example affects the model performance, and the reward function is non-differentiable as we explain below. MixRL's framework is inspired by DVRL (Yoon et al., 2020), but we solve the different problem of mixing examples and address new issues.

### 3.1 POLICY OPTIMIZATION REINFORCEMENT LEARNING

Finding the optimal policy involves taking the gradient of the objective function $J(\theta) = \mathbb{E}_{\pi_\theta}[R]$ where $\pi$ is the policy, $\theta$ is its parameters, and $R$ is the reward function. For MixRL, we would like to *minimize* the regression model loss on a small validation set when mixing a batch of examples. However, the validation loss is computed using the regression model, which does not involve $\theta$. Hence we cannot analytically compute the differential of the reward function with respect to $\theta$.

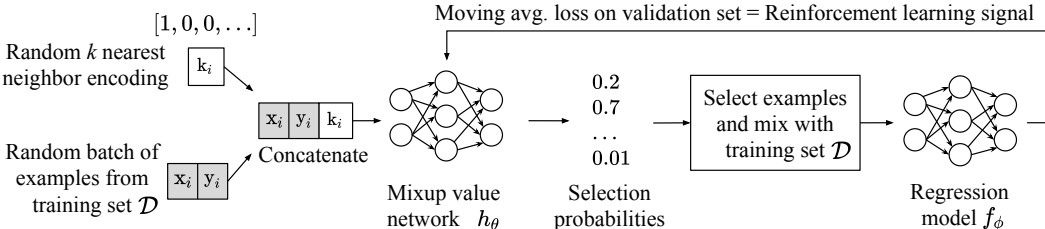

Figure 3: MixRL's framework. Given a random batch of examples from the training set, each example is combined with a one-hot encoding feature containing a random number of nearest neighbors to mix in terms of data or label distance. The Mixup value network $h_\theta$ is a neural network that returns probabilities in proportion to how helpful mixing an example with its neighbors is. The examples are then randomly selected according to the probabilities and mixed with their nearest neighbors in the training set. The random batch is augmented with the mixed examples and used to train a regression model $f_\phi$. The validation loss is used to update the Mixup value network. We also use reward scaling and employ a baseline function that takes the moving average of previous losses.

Instead, we use the REINFORCE (Williams, 1992) policy gradient algorithm, which is a Monte Carlo method that is also widely used for data valuation (Yoon et al., 2020) and neural architecture searching (Zoph & Le, 2017). It is known that $\nabla_\theta J(\theta)$ can be approximated (see Sec. A.1) as:

$$\nabla_\theta J(\theta) \approx \frac{1}{m} \sum_{i=1}^{m} R(\tau^i) \sum_{t=0}^{T-1} \nabla_\theta \log \pi_\theta(a_t^i | s_t^i) \tag{1}$$

where $\tau^i$ is the $i^{th}$ state-action trajectory under policy $\pi_\theta$, $m$ is the number of sample trajectories, $T$ is the number of actions taken in a path, $s_t^i$ is a state at time $t$, and $a_t^i$ is an action at time $t$.

In our setting of minimizing loss, REINFORCE performs gradient descent for each example decreasing $\theta$ by $\alpha \nabla_\theta J(\theta)$ where $\alpha$ is a learning rate. Although the estimated gradient is unbiased, it is known to have a high variance, which we reduce using baseline techniques (Sutton & Barto, 2018).

## 3.2 FRAMEWORK

We define notations used in MixRL's framework shown in Fig. 3. Let $\mathcal{D} = \{(x_i, y_i)\}_{i=1}^{S} \sim P$ be the training set where $x_i \in X$ is a $d$-dimensional input example, and $y_i \in Y$ is an $e$-dimensional label. Let $\mathcal{D}^v = \{(x_i^v, y_i^v)\}_{i=1}^{V} \sim P^t$ be the validation set, where $P^t$ is the distribution of the test set, which is not necessarily the same as $P$. Let $f_\phi$ be a regression model, and $L$ the loss function that returns a performance score comparing $f_\phi(x_i)$ with the true label $y_i$ using Mean Square Error (MSE). We assume a list $N$ of possible data and label nearest neighbors (NNs) that can be mixed with an example. For instance, $N$ could contain the options "1 data NN", "2 data NNs", and "2 label NNs". The more fine-grained the NN options are, the more precisely MixRL can determine the optimal number of NNs to mix per example. We do not add a "0 NN" option because selecting and not selecting it have identical effects, which makes the policy network training unstable because nothing can be learned. Instead, we support this option separately by excluding examples that are not worth mixing as we explain later in this section. The possible NNs can be represented as a one-hot encoding vector of $|N|$ values.

We now define the states and actions, which is an important design choice of MixRL. A state $s_t$ is a batch of examples $\mathcal{D}^b = \{((x_i, y_i), k_i)\}_{i=1}^{B}$ where $\{(x_i, y_i)\}_{i=1}^{B} \subseteq \mathcal{D}$ and each $k_i$ is an index of an NN option in $N$ that specifies how many data or label NNs $x_i$ should be mixed with. An action $a_t$ is then choosing $\mathcal{D}^m \subseteq \mathcal{D}^b$ where each $(x_i, y_i)$ in $\mathcal{D}^m$ is mixed with its $N[k_i]$ NNs in $\mathcal{D}$. A policy $\pi_\theta(\mathcal{D}^m | \mathcal{D}^b)$ returns the probability of selecting $\mathcal{D}^m$ at state $\mathcal{D}^b$. For each episode, MixRL selects a batch (state) once and chooses a subset of the batch to obtain a reward (action) once. Since there is only one time step, the transition function does not play a role.

A naïve implementation of the policy network would have an input dimension of $B \times (d + e + |N|)$ and an output dimension of $2^B$, which may be too large to train for large batch sizes. For example, in a typical setting of $B = 1000$, $d = 100$, $e = 1$, and $|N| = 10$, the input and output dimensions

---

**Algorithm 1:** Pseudo code for Mixup value network training.

---

**Input** : Training set $\mathcal{D}$, validation set $\mathcal{D}^v$, nearest neighbor options $N$, learning rate $\alpha$, reward scaling constant $C$, moving average window $W$
**Output:** Mixup value network $h_\theta$
Initialize $\theta$, $Base = 0$;
**while** *until convergence* **do**
    Sample $\mathcal{D}^b = \{(x_i, y_i), k_i\}_1^B \sim P \times Uniform(N)$;
    $\mathcal{D}^m = \emptyset$;
    **for** $((x_i, y_i), k_i) \in \mathcal{D}^b$ **do**
        | Add $(x_i, y_i, k_i)$ to $\mathcal{D}^m$ with probability $h_\theta(x_i, y_i, k_i)$;
    Train regression model $f_\phi$ with initialized $\phi$ on $\mathcal{D}^b \cup Mix(\mathcal{D}^m, \mathcal{D}, N)$;
    $Loss = \frac{1}{V} \sum_{i=1}^V L(f_\phi(x_i^v), y_i^v)$;
    $\theta = \theta - \alpha \cdot C \cdot (Loss - Base) \cdot \nabla_\theta \log(\pi_\theta(\mathcal{D}^m, \mathcal{D}^b))$;
    $Base = \frac{W-1}{W} Base + \frac{1}{W} Loss$;
**return** return $h_\theta$;

---

become $111,000$ and $2^{1000}$, respectively. Instead, we can significantly reduce the network size by assuming independence among examples and decomposing the policy network's prediction into two phases. First, a Mixup value network $h_\theta(x, y, k)$ is used to estimate the probability of an example being mixed with k NNs. Next, we randomly select which examples to mix according to the probabilities. The value of $\pi_\theta(\mathcal{D}^m | \mathcal{D}^b)$ is thus $\prod_{(x,y,k) \in \mathcal{D}^m} h_\theta(x, y, k) \prod_{(x,y,k) \in \mathcal{D}^b \setminus \mathcal{D}^m} [1 - h_\theta(x, y, k)]$. Also, $h_\theta$ has input and output dimensions of only $d + e + |N|$ and 1, respectively, and is practical.

The objective function $J(\theta)$ is the validation set loss $l_\theta$ of $f_\phi$ trained on $\mathcal{D}^b \cup Mix(\mathcal{D}^m, \mathcal{D}, N)$ (either from scratch or from a pre-trained state) where $Mix(\mathcal{D}^m, \mathcal{D}, N)$ mixes each $(x_i, y_i)$ in $\mathcal{D}^m$ with its $N[k_i]$ NNs in $\mathcal{D}$:

$$J(\theta) = l(\phi) = \mathop{\mathbb{E}}_{(x_i^v, y_i^v) \sim P^t} L(f_\phi(x_i^v), y_i^v) \tag{2}$$

Using Eq. 1, the gradient of Eq. 2 can be approximated using the validation set $\mathcal{D}^v$ as:

$$\nabla_\theta J(\theta) = \nabla_\theta l(\theta) \approx \frac{1}{V} \sum_{i=1}^V L(f_\phi(x_i^v), y_i^v) \cdot \nabla_\theta \log(\pi_\theta(\mathcal{D}^m, \mathcal{D}^b)) \tag{3}$$

where $\nabla_\theta \log \pi_\theta(\mathcal{D}^m, \mathcal{D}^b) = \nabla_\phi \sum_{(x,y,k) \in \mathcal{D}^m} \log h_\theta(x, y, k) + \sum_{(x,y,k) \in \mathcal{D}^b \setminus \mathcal{D}^m} [1 - h_\theta(x, y, k)]$. We also use reward scaling (Henderson et al., 2018) to further improve the training.

Algorithm 1 shows the pseudo code for training $h_\theta$. The computational complexity is not directly related to the training set size, but depends on the number of iterations needed to train $h_\theta$ and how long each iteration takes. Another factor is the number of possible NNs $|N|$ where a higher number may result in slower training. We show in Sec. 4.2 that MixRL scales to large training sets.

Once we train $h_\theta$, we choose the $(x, y, k)$'s with the highest $h_\theta(x, y, k)$ values of at least a threshold $T$. We optimize $T$'s value using the validation set (see Sec. B.1). We ignore any $(x, y, k)$ whose $(x, y)$ pair has already been chosen with a different k value for the same data or label distance. We apply $T$ in order to exclude $(x, y, k)$'s that are not worth mixing (i.e., make them mix with 0 NNs).

Although actor-critic methods (Mnih et al., 2016; Schulman et al., 2017) improve on REINFORCE by training both value and policy networks, using the two networks may not be practical in our setting due to the high input dimensions. For a policy network, we can reduce its dimension using the fact that the log probabilities can be calculated by multiplying selection probabilities. Unfortunately, we cannot use the same trick for a value network because its output values are not probabilities.

## 4 EXPERIMENTS

We provide experimental results for MixRL. We evaluate the regression models trained on the augmented training sets on separate test sets and repeat all model trainings 5 times. We use Py-Torch (Paszke et al., 2017), and all experiments are performed using Intel Xeon Silver CPUs and NVidia RTX GPUs. More experimental setting details are in Sec. B.1.

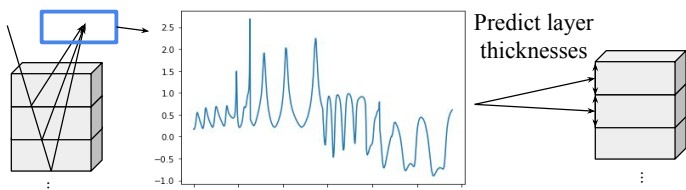

Figure 4: Spectrum generation on a 3-d semiconductor for the Product dataset.

Table 1: Settings for the four datasets.

| Dataset | Data dim. | Label dim. | Training set size | Val. set size | Test set size |
|---|---|---|---|---|---|
| NO2 | 7 | 1 | 200 | 200 | 100 |
| Product | 580 | 20 | 300 | 300 | 200 |
| Synthetic | 226 | 4 | 100 | 100 | 100 |
| Airfoil | 5 | 1 | 1003 | 300 | 200 |

**Measures** We use two accuracy metrics. RMSE = $\sqrt{\frac{1}{n}\sum_{i=1}^{n}(y_i - \hat{y}_i)^2}$ is the absolute difference between predicted labels and true labels where a lower value is better. $R^2 = 1 - \frac{\sum_{i=1}^{n}(y_i - \hat{y}_i)^2}{\sum_{i=1}^{n}(y_i - \bar{y})^2}$ is the relative improvement compared to when returning the label average as a prediction. The value is in the range [0, 1] where a higher value is better. The two measures complement each other.

**Datasets** We use three real and one synthetic datasets. The NO2 emissions dataset (Aldrin, 2004) contains traffic and meteorological information around roads. The features include cars per hour, wind speed, temperature, and others. The label is the NO2 concentration. The Product dataset obtained through a collaboration with a company contains spectrum data that is generated using the procedure described in Sec. 1 and illustrated in Fig. 4 on 20-layer 3-d semiconductors. The Product dataset is proprietary, so for clarity we also experiment on a public synthetic dataset (Synthetic) provided by the DACON challenge (DACON Co., Ltd., 2020) where a simulator generates spectrum data using the same procedure as Product assuming 4-layer 3-d semiconductors. Finally, the Airfoil UCI dataset (Dua & Graff, 2017) contains aerodynamic and acoustic test results for airfoil blade sections in a wind tunnel where the features include frequency, angle of attack, and chord length. Table 1 compares the four datasets in more detail. Note that Product and Synthetic have high dimensions, while Airfoil has the largest training set size.

**MixRL Settings** For the regression model, we use a multi-layer perceptron (MLP) with 2 to 4 hidden layers and set the number of nodes per hidden layer to be [512, 256] for NO2, [2048, 1024, 512, 256] for Product, and [2048, 1024, 512, 256] for Synthetic. We employ layer normalization, early stopping for regularization, and reward scaling and baseline techniques that improve the performance and stability of reinforcement learning. We use the Adam optimizer for all MLP trainings. When mixing examples, we set the Mixup ratio $\lambda$ to 0.5 unless specified otherwise. This value empirically results in the best Mixup performance in a regression setting, and we give an explanation in Sec. B.3. For the possible numbers of NNs for NO2, our default setting is the series of integers $\{4, 16, 64\}$ for both data and label distances, i.e., $|N| = 6$. For Product, Synthetic, and Airfoil, the default series are $\{100, 150, 200, 250\}$, $\{4, 16, 64\}$, and $\{1, 2, 4\}$, respectively.

**Baselines** We employ four baselines. First, we train a regression model on the labeled data without any data augmentation ("No Augmentation"). Next, we faithfully implement the original Mixup algorithm (Zhang et al., 2018) where all example pairs can be mixed ("Original Mixup"). We also implement Manifold Mixup (Verma et al., 2019), which is a state-of-the-art Mixup technique for classification that is also applicable to regression. Finally, we compare with a simplified version of MixRL where we use a single k NN value to mix all examples with their data and label neighbors ("Global kNN"). This k parameter is tuned using Bayesian Optimization (Mockus, 1974).

### 4.1 Limited Linearity in Regression

We empirically show that the linearity assumption does not hold from certain data or label distances using the following simple experiment. Given a dataset, we first train a regression model. We then

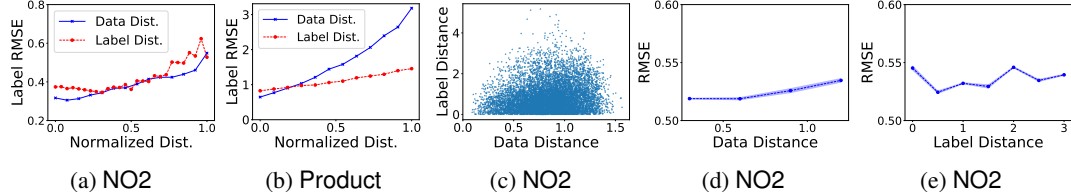

(a) NO2      (b) Product      (c) NO2      (d) NO2      (e) NO2

Figure 5: (a) For NO2, the label RMSE increases for larger (normalized) data or label distances. (b) Similar label RMSE trends for Product. (c) For NO2, the data and label distances do not have a linear relationship. (d) For NO2, the regression model's RMSE increases for larger data distances. (e) For NO2, the model's RMSE decreases, but then increases for larger label distances.

Table 2: Accuracy and runtime results on the NO2, Product, and Airfoil datasets. Four baselines are compared with MixRL: (1) No data augmentation; (2) Original Mixup (Zhang et al., 2018); (3) Manifold Mixup (Verma et al., 2019); and (4) Global kNN. For each global kNN, we show the best k value found with Bayesian optimization. We also integrate MixRL with Manifold Mixup.

| Dataset | Method | RMSE | $R^2$ | Runtime (mins) |
|---|---|---|---|---|
| NO2 | No Augmentation | $0.5123_{\pm 0.0009}$ | $0.5128_{\pm 0.0008}$ | 0.01 |
| | Original Mixup | $0.5152_{\pm 0.0015}$ | $0.5018_{\pm 0.0025}$ | 0.01 |
| | Manifold Mixup | $0.5077_{\pm 0.0024}$ | $0.5182_{\pm 0.0046}$ | 0.02 |
| | Global kNN (k=64) | $0.5089_{\pm 0.0027}$ | $0.5149_{\pm 0.0040}$ | 0.5 |
| | MixRL | $\mathbf{0.4937}_{\pm 0.0046}$ | $\mathbf{0.5445}_{\pm 0.0068}$ | 40 |
| | MixRL + Manifold Mixup | $0.5061_{\pm 0.0041}$ | $0.5221_{\pm 0.0061}$ | 45 |
| Product | No Augmentation | $1.3525_{\pm 0.0068}$ | $0.7208_{\pm 0.0028}$ | 2 |
| | Original Mixup | $1.1617_{\pm 0.0084}$ | $0.7884_{\pm 0.0034}$ | 2 |
| | Manifold Mixup | $1.2553_{\pm 0.0086}$ | $0.7490_{\pm 0.0030}$ | 3 |
| | Global kNN (k=267) | $1.1575_{\pm 0.0101}$ | $0.7846_{\pm 0.0037}$ | 20 |
| | MixRL | $\mathbf{1.1383}_{\pm 0.0057}$ | $\mathbf{0.7924}_{\pm 0.0020}$ | 390 |
| | MixRL + Manifold Mixup | $1.2475_{\pm 0.0124}$ | $0.7509_{\pm 0.0045}$ | 400 |
| Airfoil | No Augmentation | $1.2924_{\pm 0.0279}$ | $0.9650_{\pm 0.0015}$ | 3 |
| | Original Mixup | $2.6394_{\pm 0.0251}$ | $0.8665_{\pm 0.0027}$ | 3 |
| | Manifold Mixup | $1.4495_{\pm 0.0056}$ | $0.9567_{\pm 0.0006}$ | 11 |
| | Global kNN (k=1) | $1.5072_{\pm 0.0259}$ | $0.9520_{\pm 0.0016}$ | 25 |
| | MixRL | $1.2346_{\pm 0.0325}$ | $0.9681_{\pm 0.0015}$ | 410 |
| | MixRL + Manifold Mixup | $\mathbf{1.2030}_{\pm 0.0311}$ | $\mathbf{0.9697}_{\pm 0.0015}$ | 460 |

generate mixed examples and compute the RMSE between their interpolated labels and the labels predicted by the regression model. While the regression model is not perfect, the relative trend of the error gives sufficient insight. Figs. 5a and 5b indeed show increasing label errors for larger label and data distances for the NO2 and Product datasets, respectively. Figs. 2c and 5c also show how label and data distances do not necessarily have a linear relationship for the two datasets. We also train regression models on top of augmented training sets where we only mix examples within certain distance ranges. As a result, mixing examples with larger data or label distances has increasingly-negative effects on the model accuracy as shown in Figs. 2a, 2b, 5d, and 5e.

## 4.2 ACCURACY AND RUNTIME RESULTS

We compare MixRL with the four baselines using the settings in Table 1. Table 2 shows the results for the three real datasets (the Synthetic dataset results are in Sec. B.2). MixRL significantly outperforms all the baselines in terms of regression model accuracy, which demonstrates that it is able to pinpoint which examples should be mixed with how many kNNs. Figs. 6a and 6b show which kNN options are frequently used for NO2 and Product (the kNN options for Synthetic and Airfoil are in Sec. B.2). Although the runtime is slower than the baselines due to the reinforcement learning, the gained accuracy can be worth the effort in real applications because data augmentation is a one-time cost that can run in batch mode. Another observation is that Original Mixup performs relatively bet-

Figure 6: (a)-(b) kNN option frequency histograms for two real datasets. (c)-(d) Model accuracy vs. validation set size for two real datasets. (e) MixRL's runtime against the training set size for Airfoil.

Table 3: Comparing kNN options for MixRL on the NO2 and Product datasets.

| Dataset | Data and Label kNN Options | RMSE | $R^2$ |
|---|---|---|---|
| NO2 | $\{4, 16\}$ | $0.4994_{\pm 0.0045}$ | $0.5305_{\pm 0.0077}$ |
| | $\{4, 16, 64\}$ (Default) | $0.4937_{\pm 0.0046}$ | $0.5445_{\pm 0.0068}$ |
| | $\{4, 16, 64, 128\}$ | $\mathbf{0.4932}_{\pm 0.0023}$ | $\mathbf{0.5447}_{\pm 0.0032}$ |
| Product | $\{100, 150\}$ | $1.1484_{\pm 0.0040}$ | $0.7902_{\pm 0.0009}$ |
| | $\{100, 150, 200, 250\}$ (Default) | $1.1383_{\pm 0.0057}$ | $0.7924_{\pm 0.0020}$ |
| | $\{100, 150, 200, 250, 300\}$ | $\mathbf{1.1372}_{\pm 0.0046}$ | $\mathbf{0.7942}_{\pm 0.0015}$ |

ter for Product than other datasets because linearity holds more as evidenced by the large k = 267 value of Global kNN. In general, MixRL outperforms Original Mixup by a larger margin the more limited the linearity. We also integrate MixRL with Manifold Mixup where examples are mixed in multiple layers of the MLP with their NNs (see Sec. B.4 for more details). The results are mixed where the integrated method performs the best on the Airfoil dataset, but not on the other datasets.

**Validation Set and Scalability** Figs. 6c and 6d show the validation set size impact on MixRL's accuracy for the NO2 and Product datasets. The Airfoil dataset results are similar and shown in Sec. B.2. We observe that a small validation set is sufficient for MixRL to be effective. Also MixRL scales well against increasing training set sizes as shown in Fig. 6e.

### 4.3 K NEAREST NEIGHBOR OPTIONS

We investigate how changing the kNN options $N$ affects MixRL's performance in Table 3 using the NO2 and Product datasets. The other dataset results are similar and shown in Sec. B.2. For each dataset, we compare its default series with shorter and longer series. We observe that providing more kNN options improves MixRL's performance, but only to a certain extent. Hence, MixRL performs well even with a small number of options.

### 4.4 ABLATION STUDY

We perform an ablation study to investigate the effect of limiting label and data distances in Table 4 using the NO2 and Product datasets. The other dataset results are similar and shown in Sec. B.2. We compare MixRL with three variants: (1) MixRL that does not exploit any distances and selects examples that have the highest $h_\theta$ values when fully mixed with all the other examples; (2) MixRL that does not use label distances; and (3) MixRL that does not use data distances. We observe that using both distances is necessary for the best accuracy results as we suggested in Sec. 2.

## 5 RELATED WORK

**Data Augmentation for Regression** There are largely two branches of work for data augmentation in regression. One is semi-supervised regression (Kostopoulos et al., 2018; Zhou & Li, 2005; Kang et al., 2016; Kim et al., 2020) where the goal is to utilize unlabeled data for training. In comparison, MixRL does not assume the availability of unlabeled data. Another branch is data augmentation when there is no unlabeled data, which is our research focus. To the best of our knowledge, there is no technique tailored to regression, so we instead cover data augmentation techniques for classification that can be extended to regression.

Table 4: Ablation study for MixRL on the NO2 and Product datasets. Three variants are compared: (1) w/o label and data distance limits; (2) w/o label distance limits; and (3) w/o data distance limits.

| Dataset | Method | RMSE | $R^2$ |
|---|---|---|---|
| NO2 | W/O Label & Data Dist. Limits | $0.5158_{\pm 0.0026}$ | $0.5019_{\pm 0.0035}$ |
| | W/O Label Dist. Limits | $0.5007_{\pm 0.0017}$ | $0.5298_{\pm 0.0028}$ |
| | W/O Data Dist. Limits | $0.5026_{\pm 0.0045}$ | $0.5331_{\pm 0.0087}$ |
| | MixRL | $\mathbf{0.4937}_{\pm 0.0046}$ | $\mathbf{0.5445}_{\pm 0.0068}$ |
| Product | W/O Label & Data Dist. Limits | $1.1561_{\pm 0.0103}$ | $0.7858_{\pm 0.0041}$ |
| | W/O Label Dist. Limits | $1.1652_{\pm 0.0074}$ | $0.7827_{\pm 0.0030}$ |
| | W/O Data Dist. Limits | $1.1471_{\pm 0.0048}$ | $0.7889_{\pm 0.0022}$ |
| | MixRL | $\mathbf{1.1383}_{\pm 0.0057}$ | $\mathbf{0.7924}_{\pm 0.0020}$ |

We explore data augmentation techniques for classification and investigate if they can be used in a regression setting. There are largely three approaches: generative models, policies, and Mixup techniques. Generative models including GANs (Goodfellow et al., 2014) and VAEs (Kingma & Welling, 2014) are popular in classification where the idea is to generate realistic data that cannot be distinguished from the real data by a discriminator. However, a major assumption is that the labels of the generated examples are the same, which does not necessarily hold in a regression setting where most examples may have different labels. Another approach is to use policies (Cubuk et al., 2019), which specify fixing rules for transforming the data while maintaining the label value. For example, image processing policies may flip, rotate, or adjust the brightness of images to generate new valid images. However, in a regression setting, the same transformed examples may now have different unknown labels, making them unsuitable for training. The recently-proposed Mixup (Zhang et al., 2018; Berthelot et al., 2019; Yun et al., 2019; Berthelot et al., 2020; Hendrycks et al., 2020; Verma et al., 2019) takes the alternative approach of generating both data and labels together by mixing existing examples with different labels assuming linearity between training examples (Chapelle et al., 2000; Wu et al., 2020). Although the Mixup paper mentions that its techniques can easily be extended to regression, we make the non-trivial observation that the linearity assumption does not necessarily hold where mixing examples with far-away examples may not be beneficial and even detrimental to model performance. Instead, MixRL shows good performance by carefully choosing which nearest neighbors to mix with, a technique that only works in a regression setting.

**Policy Optimization Reinforcement Learning** Within reinforcement learning (RL) (Sutton & Barto, 2018; Kaelbling et al., 1996), we are interested in policy searching methods (Neumann & Peters, 2015) where our goal is to find the best policy for mixing examples. In particular, MixRL's framework is inspired by the recent Data Valuation using Reinforcement Learning (DVRL) framework (Yoon et al., 2020), which uses REINFORCE (Williams, 1992) to estimate the value of data in terms of how much it improves the model performance. In comparison, we solve the completely different problem of data augmentation for regression where we extend Mixup using REINFORCE. Some data augmentation techniques use actor-critic methods like A3C (Mnih et al., 2016) and PPO (Schulman et al., 2017) that train value networks along with policy networks. For example, AutoAugment (Cubuk et al., 2019) uses PPO to search for data augmentation policies that dictate how to modify existing examples to generate additional training data for image classification. However, using actor-critic methods is practically challenging in our setting due to the large value networks that need to be trained as we discussed in Sec. 3.2.

## 6 CONCLUSION

We proposed MixRL, which is to our knowledge the first Mixup-based data augmentation meta learning framework tailored to regression tasks. We observe that linearity only holds up to a certain data or label distance per example in a regression setting. Hence, MixRL extends Mixup by determining the k nearest neighbors to mix per example using Monte Carlo policy gradient reinforcement learning where the objective is to minimize the regression model's loss on a small validation set. Since MixRL only limits the examples to mix, it can be integrated with any existing Mixup technique for classification. In our experiments, we showed that MixRL outperforms various data augmentation baselines for regression by effectively selecting the k nearest neighbor options.

**Ethics Statement**  MixRL demonstrates that data augmentation can significantly enhance the model accuracy for regression tasks. However, if not controlled, data augmentation may also introduce new bias in the data that makes the model more discriminative. Thus an interesting future work is to augment data while ensuring model fairness.

**Reproducibility Statement**  We add MixRL's source code in the supplementary. We also provide more details on Eq. 1, experimental settings, and experimental results in the appendix.

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

## A    APPENDIX – FRAMEWORK

### A.1    GRADIENT OF $J(\theta)$

We continue from Sec. 3.1 and derive Eq. 1 (Brunskill, 2021):

$$\nabla_\theta J(\theta) \approx \frac{1}{m} \sum_{i=1}^{m} R(\tau^i) \sum_{t=0}^{T-1} \nabla_\theta \log \pi_\theta(a_t^i | s_t^i) \tag{1}$$

where $\tau_i$ is a state-action trajectory under policy $\pi_\theta$, $m$ is the number of sample trajectories, $T$ is the number of actions taken in a path, $s_t$ is a state at time $t$, and $a_t$ is an action at time $t$. Suppose that $P(\tau; \theta)$ is the probability of a trajectory $\tau$ when using the policy $\pi_\theta$, and $R(\tau)$ is the reward of $\tau$. Then $J(\theta)$ can be expressed as $\sum_\tau P(\tau; \theta) R(\tau)$. In addition, let us specify $s_0$ as the initial state, $\mu(s_0)$ as the initial state's distribution, and $P(s_{t+1}|s_t, a_t)$ as the dynamics model that specifies which predicts the next state $s_{t+1}$ when taking action $a_t$ from state $s_t$. We can then take the gradient of $J(\theta)$ as follows:

$$
\begin{aligned}
\nabla_\theta J(\theta) =& \nabla_\theta \sum_\tau P(\tau; \theta) R(\tau) \\
=& \sum_\tau \nabla_\theta P(\tau; \theta) R(\tau) \\
=& \sum_\tau \frac{P(\tau; \theta)}{P(\tau; \theta)} \nabla_\theta P(\tau; \theta) R(\tau) \\
=& \sum_\tau P(\tau; \theta) R(\tau) \frac{\nabla_\theta P(\tau; \theta)}{P(\tau, \theta)} \\
=& \sum_\tau P(\tau; \theta) R(\tau) \nabla_\theta \log P(\tau; \theta) \\
\approx& \frac{1}{m} \sum_{i=1}^{m} R(\tau^i) \nabla_\theta \log P(\tau^i; \theta) \\
=& \frac{1}{m} \sum_{i=1}^{m} R(\tau^i) \nabla_\theta \log \left[ \mu(s_0) \prod_{t=0}^{T-1} \pi_\theta(a_t^i | s_t^i) P(s_{t+1}^i | s_t^i, a_t^i) \right] \\
=& \frac{1}{m} \sum_{i=1}^{m} R(\tau^i) \nabla_\theta \left[ \log \mu(s_0^i) \sum_{t=0}^{T-1} \log \pi_\theta(a_t^i | s_t^i) + \log P(s_{t+1}|s_t^i, a_t^i) \right] \\
=& \frac{1}{m} \sum_{i=1}^{m} R(\tau^i) \sum_{t=0}^{T-1} \nabla_\theta \log \pi_\theta(a_t^i | s_t^i)
\end{aligned}
$$

## B    APPENDIX – EXPERIMENTS

### B.1    MORE EXPERIMENTAL SETTINGS

We continue describing our experimental settings from Sec. 4. For the regression model, we use layer normalization (Ba et al., 2016) for the Product, Synthetic, and Airfoil datasets. We also use early stopping to prevent overfitting on the validation set for all the datasets. For the mixup value network $h_\theta$, we use reward scaling and baseline techniques that improve the performance and stability of reinforcement learning. For the mixup value network $h_\theta$, we use a multi-layer perceptron with 4 hidden layers and set the number of nodes per hidden layer to be [100, 100, 100, 100] for the four datasets. For $h_\theta$, we use a learning rate of $\alpha = 0.001$ for all datasets. For the regression model $f_\phi$, we use a learning rate of 0.0001 for Product and 0.001 for the other datasets. When training $h_\theta$, we set the batch size $B$ to be the training data's size. When training $f_\phi$, we use a batch size of 32 for NO2, 64 for Product, 32 for Synthetic, and 64 for Airfoil. We use a reward scaling (Henderson et al., 2018) constant of $C = 5$ for NO2, $C = 1$ for Product, $C = 0.1$ for Synthetic, and $C = 1$ for

Airfoil. We use a moving average window of size $W = 20$ for the three datasets. When setting the threshold $T$ for limiting examples to mix, we choose a value that results in a high model accuracy (low RMSE) on the validation set.

When constructing the Synthetic dataset, we take a subset of the entire DACON challenge dataset (DACON Co., Ltd., 2020). While the the range of thicknesses for the full datasets is [10, 300] for all four layers, we only use examples with thicknesses within the range [10, 50]. The purpose is to reduce the training time while still making a clear comparison between MixRL and the other methods.

To improve the runtime of MixRL, we use pre-trained regression models for the Product, Synthetic, and Airfoil datasets. First, we train the base regression model on the training data. Then, we fine-tune the pre-trained model to obtain the validation losses.

### B.2 MORE Synthetic AND Airfoil DATASET EXPERIMENTS

**Accuracy and Runtime Results**    We continue from Sec. 4.2 and show the accuracy and runtime results for the Synthetic dataset in Table 5. The results are similar to the Product dataset in Table 2.

Table 5: Accuracy and runtime performances for the Synthetic dataset. The other settings are identical to those of Table 2.

| Dataset | Method | RMSE | $R^2$ | Runtime (mins) |
|---------|--------|------|-------|----------------|
| Synthetic | No Augmentation | $9.2624_{\pm 0.0472}$ | $0.5742_{\pm 0.0010}$ | 0.1 |
| | Original Mixup | $8.4932_{\pm 0.0532}$ | $0.6357_{\pm 0.0034}$ | 0.1 |
| | Manifold Mixup | $8.9210_{\pm 0.0687}$ | $0.5973_{\pm 0.0066}$ | 0.1 |
| | Global kNN (k=43) | $8.4113_{\pm 0.0573}$ | $0.6400_{\pm 0.0039}$ | 11 |
| | MixRL | $\mathbf{8.3567}_{\pm 0.0339}$ | $\mathbf{0.6463}_{\pm 0.0043}$ | 260 |
| | MixRL + Manifold Mixup | $8.5958_{\pm 0.0679}$ | $0.6297_{\pm 0.0047}$ | 370 |

**kNN Options Frequency Histogram**    We continue from Sec. 4.2 and show which kNN options are frequently used in our default settings for the Synthetic and Airfoil datasets in Figs. 7a and 7b, respectively. Compared to the Product dataset, Synthetic has no cases where $k = 0$ because mixing is more beneficial to the model's accuracy according to the validation set.

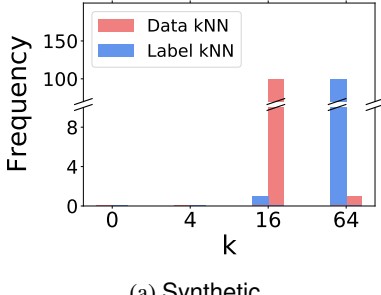
(a) Synthetic

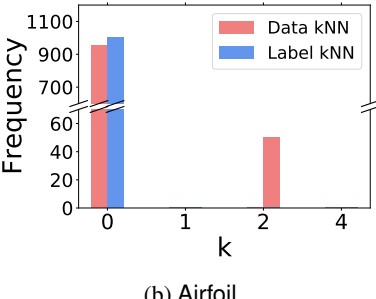
(b) Airfoil

Figure 7: kNN option frequency histograms for the (a) Synthetic dataset and (b) Airfoil dataset.

**Validation Set Size and Accuracy**    We continue from Sec. 4.2 and show how the validation set size impacts MixRL's accuracy for the Airfoil dataset in Fig. 8. The observations are similar to those of the other datasets where a small validation set size is sufficient for MixRL to perform well.

**kNN Options Comparison**    We continue from Sec. 4.3 and provide the kNN options experiments for the Synthetic and Airfoil datasets in Table 6. The observations are similar to those of Table 3 where a small number of options is sufficient for good model accuracy.

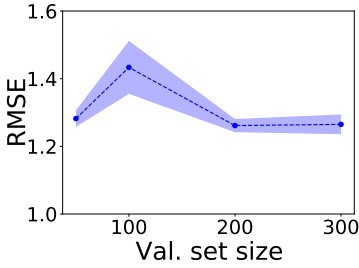

Figure 8: Model accuracy vs. validation set size for the Airfoil dataset.

Table 6: Comparing kNN options for MixRL on the Synthetic and Airfoil datasets.

| Dataset | Data and Label kNN Options | RMSE | $R^2$ |
|---|---|---|---|
| Synthetic | $\{4, 16\}$ | $8.9018_{\pm 0.0490}$ | $0.5954_{\pm 0.0034}$ |
| | $\{4, 16, 64\}$ (Default) | $\mathbf{8.3567}_{\pm 0.0339}$ | $\mathbf{0.6463}_{\pm 0.0043}$ |
| | $\{4, 16, 64, 100\}$ | $8.5211_{\pm 0.0214}$ | $0.6291_{\pm 0.0027}$ |
| Airfoil | $\{1, 2\}$ | $1.2824_{\pm 0.0201}$ | $0.9660_{\pm 0.0010}$ |
| | $\{1, 2, 4\}$ (Default) | $\mathbf{1.2346}_{\pm 0.0325}$ | $\mathbf{0.9681}_{\pm 0.0015}$ |
| | $\{1, 2, 4, 8\}$ | $1.2546_{\pm 0.0096}$ | $0.9678_{\pm 0.0001}$ |

**Ablation Study** We continue from Sec. 4.4 and provide the ablation study for the Synthetic and Airfoil datasets in Table 7. The results are similar to those in Table 4 where using both label and data distances results in the best model accuracies.

Table 7: Ablation study for MixRL on the Synthetic and Airfoil datasets. Three variants compared: (1) w/o label and data distance limits; (2) w/o label distance limits; and (3) w/o data distance limits.

| Dataset | Method | RMSE | $R^2$ |
|---|---|---|---|
| Synthetic | W/O Label & Data Dist. Limits | $8.4392_{\pm 0.0658}$ | $0.6370_{\pm 0.0068}$ |
| | W/O Label Dist. Limits | $9.2794_{\pm 0.0554}$ | $0.5708_{\pm 0.0037}$ |
| | W/O Data Dist. Limits | $8.3720_{\pm 0.0396}$ | $0.6424_{\pm 0.0032}$ |
| | MixRL | $\mathbf{8.3567}_{\pm 0.0339}$ | $\mathbf{0.6463}_{\pm 0.0043}$ |
| Airfoil | W/O Label & Data Dist. Limits | $1.2852_{\pm 0.0184}$ | $0.9659_{\pm 0.0009}$ |
| | W/O Label Dist. Limits | $1.2447_{\pm 0.0250}$ | $0.9678_{\pm 0.0013}$ |
| | W/O Data Dist. Limits | $1.2552_{\pm 0.0118}$ | $\mathbf{0.9682}_{\pm 0.0006}$ |
| | MixRL | $\mathbf{1.2346}_{\pm 0.0325}$ | $0.9681_{\pm 0.0015}$ |

### B.3 JUSTIFICATION FOR USING $\lambda = 0.5$

We continue from Sec. 4 and provide more details on why setting $\lambda = 0.5$ empirically results in the best Mixup performance in a regression setting. Figure 9 shows MixRL's accuracy when we vary $\lambda$ using the same setting in Table 1 for the NO2 and Product datasets. As a result, the accuracy peaks when $\lambda = 0.5$. We suspect that generating examples in the sparsest areas helps the regularization the most. In a classification setting, however, this strategy may not be ideal especially if the generated examples are too close to a decision boundary that is in the middle of the mixed examples.

### B.4 INTEGRATING MIXRL WITH MANIFOLD MIXUP

We continue from Sec. 4.2 and provide more details on integrating MixRL with Manifold Mixup (Verma et al., 2019). We first run MixRL and determine which examples to mix. Next, instead of mixing examples in the training data before model training, we mix examples on multiple layers in the MLP during model training. For the second mixing step, we use Manifold Mixup as is

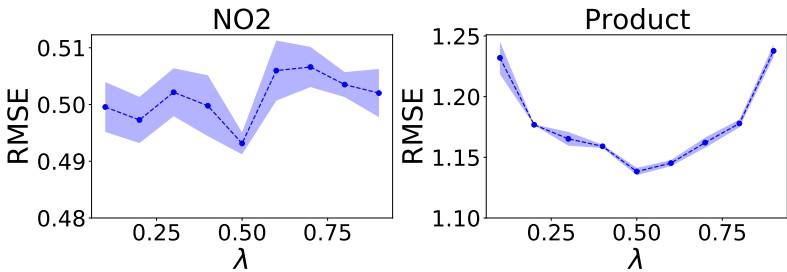

Figure 9: MixRL's accuracy for different $\lambda$ values on the NO2 and Product datasets.

where we start with a set of eligible layers for mixing. (In our experiments, we always set the first three layers as eligible.) For each training batch, we randomly select a mixing layer and mix the features and labels of the examples that MixRL decided to mix.

