# OpenReview forum: "MixRL: Data Mixing Augmentation for Regression using Reinforcement Learning"
_ICLR.cc/2022/Conference — ICLR 2022 Submitted_

### Official Review · Reviewer_3Wyg · 2021-11-03

**Correctness:** 3
**Technical Novelty And Significance:** 3
**Empirical Novelty And Significance:** 3
**Recommendation:** 5
**Confidence:** 3

**Main Review:**

MixRL is inspired by the problem of measuring how individual examples contribute to model performance. The idea is reasonable for regression tasks.
The experiments verify its effectiveness by improving regression performance by carefully mixing examples.
The paper is well written and organized.

I notice MixRL cannot improve previous Mixup methods by a clear margin. Since I am hesitant about the importance of the improvement, I want to hear about other reviewers’ opinions to make the decision.

**Summary Of The Paper:**

To apply Mixup for regression tasks, the paper first utilizes the stricter assumption that linearity only holds within specific data or label distances for regression.
Then this paper proposes a data mixing augmentation method called MixRL. The goal of MixRL is to identify which examples to mix with which nearest neighbors. MixRL employs a meta-learning framework that estimates how important mixing a sample is to minimize the model loss on a validation set using policy gradient reinforcement learning.

**Summary Of The Review:**

I'd like to see the other reviewers' opinions about the empirical results.

---

> ### Author Response · Authors · 2021-11-16
> **Response to Reviewer 3Wyg**
>
> We appreciate the positive comments. We agree with the comment on the improvements and would like to make two points. First, MixRL *consistently* outperforms baselines in all settings. In comparison, the original Mixup (Zhang 2018) performs reasonably well on the Product dataset (although not as much as MixRL), but worse than when using no augmentation on the NO2 and Airfoil datasets. Second, a small, but solid improvement in accuracy can be a big deal in large companies that make say billions of dollars in revenue.

---

### Official Review · Reviewer_aYea · 2021-11-04

**Correctness:** 3
**Technical Novelty And Significance:** 2
**Empirical Novelty And Significance:** 2
**Recommendation:** 5
**Confidence:** 4

**Main Review:**

Strengths:

-Data-dependent input/output proximity constraints on mixup are learned.

-Consistent but small gains over mixup and manifold mixup are realized on several regression datasets.

Limitations:

-The set of nearest neighbors considered for use in mixup, while dynamic, are highly restricted (e.g. {4, 16, 64, 128} for the NO2 dataset in table 3.

-More generally, the lack of a local input/output kernel to prioritize more local data in a continuous manner feels like a significant limitation.

-Optimized input/output kernels that restrict and more generally re-weight mixup pairs or construct locally non-constant regressions to sample from feel like important, missing baselines. The lack of a locally non-constant regression model in MixRL (beyond mixing data pairs) is likely limiting performance significantly.

**Summary Of The Paper:**

The authors propose MixRL to improve upon mixup in regression settings. MixRL is used to impose a proximity constraint on the input/output pairs that are mixed during mixup-based data augmentation, by predicting how many nearest neighbors to utilize, from a small set of pre-specified options, based on feedback from evaluating the validation set. Consistent but small gains over mixup and manifold mixup are realized on several datasets.

**Summary Of The Review:**

The technique while sound is currently quite limited in scope in that important baselines and algorithmic elements (e.g. locally non-constant models and local kernels) that need to be considered in the context of doing high fidelity data augmentation for regression problems have not been adequately investigated or discussed.

---

> ### Author Response · Authors · 2021-11-16
> **Response to Reviewer aYea**
>
> We appreciate the insightful comments and will reflect them in our revision.
>
> Comment 1: Restricted kNN options
>
> Thanks for raising this point. As shown in Table 3 in our paper, adding more kNN options did not significantly improve accuracy, so there are diminishing returns. Adding options unnecessarily would only result in longer runtimes for finding the best options. We will clarify these points in our revision.
>
> Comment 2: Using local input/output kernels
>
> Thanks for the great suggestion. We agree using input/output kernels to re-weight mixup pairs can help avoid unnecessary mixing. On the other hand, there is an added complexity of tuning the kernels themselves. If not done properly, a kernel may reduce the weights of distant mixup pairs too much even if they should be mixed according to a kNN option. In addition, it is not clear if a single kernel setup suffices for all mixup pairs or whether there must be a different kernel setup per example with its nearest neighbors. We will add experiments and a discussion in our revision.

---

### Official Review · Reviewer_XF4Y · 2021-11-04

**Correctness:** 2
**Technical Novelty And Significance:** 2
**Empirical Novelty And Significance:** 2
**Recommendation:** 3
**Confidence:** 4

**Main Review:**

Strengths:
The idea is simple and appears to be implemented correctly, and the experiments appear to be justifiable given mixup is a useful regularization technique for supervised learning, and has been shown to be very useful in classification tasks. The application of REINFORCE as a stochastic gradient approximator is well known in other context that involve NNs with discrete processes (see DLGMs, e.g., Rezende 2014). The experiments are straightforward and support the value of their work.

Weaknesses:
I have deep concerns about this paper being marketed as "RL for mixup". RL comes packaged with a great deal more than just the REINFORCE algorithm, e.g., the policy gradient, and rephrasing a lot of what is a gradient estimator for a discrete process as "RL" not only may potentially confuses the reader on what RL is, but also doesn't connect with an active area of research on doing backprop through discrete processes. Yes, I understand that the cited Yoon 2020 paper did this as well, but I don't think they should have either. One of the most famous examples of using REINFORCE is with discrete VAEs or DLGMs (Rezende 2014), and they are careful not to call it "RL"

But there are a number of other works or baselines that this paper should have compared to, and I think in part this is because the related works talk about other RL algorithms: for instance A3C or even PPO aren't really relatable to the problem being addressed here as straight-through estimators (Bengio 2013) or Gumbel Softmax (Jang 2017) (Yoon does a better job at connecting to these types of works).

For instance, it seems possible to backprop through to the selection probabilities using Gumbel Softmax. Variance is usually much better than REINFORCE, even with the baseline removal.

These are important baselines and it is important this paper places itself correctly among related works.

Next, there are many claims about why classification doesn't need this sort of mixup, but I didn't follow the arguments entirely (P1-P2 in Section 2). I don't see any classification experiments to back up this claim, at least in the main text.  For instance "Linear assumption (in classification) turns out to be "reasonable" because the label difference..." Why is it reasonable to leave the label space in classification mixup: with regression I have a better chance of hitting a "label" in the distribution.

Other notes / concerns:
So the motivation was on some physical systems, and we're using REINFORCE to learn how to mix data. But if we're in the physical systems such as you describe, usually we have some sort of (closed-form solution) model that describes the physical process (maybe poorly). Even if this model has error, wouldn't this be a good place to design a prior that tells you how to sample? e.g., if the masses are some set distance apart, then use, else don't.

I see meta-learning as a description of the model, but not in the algorithm 1.

How do you sample k when you decide how many neighbors to mix? Is this sampled from a prior?

Rezende 2014: Stochastic Backpropagation and Approximate Inference in Deep Generative Models
Bengio 2013: Estimating or Propagating Gradients Through Stochastic Neurons for Conditional Computation
Jang 2017: Categorical Reparameterization with Gumbel-Softmax

**Summary Of The Paper:**

This paper improves on the idea of mixup by selecting samples for mixup via a model that selects suitable pairs found using knn in a batch. In order to provide a learning signal to this discrete process, the authors apply REINFORCE, using the loss of the downstream regressor (not applied to classification tasks). They show promising results on a number of regression tasks.

**Summary Of The Review:**

I recommend reject, as the paper's core story isn't well placed compared to related works. The paper is placed as "RL" so we're missing important baselines and connections to other gradient estimators through discrete processes.

---

> ### Author Response · Authors · 2021-11-16
> **Response to Reviewer XF4Y**
>
> We appreciate the thoughtful comments, which we will address in our revision.
>
> Comment 1: Concerns for RL
>
> We appreciate the great advice. We agree we may have overemphasized RL, but believe our key contribution of solving data augmentation for regression remains intact. We first clarify why we used REINFORCE instead of stochastic gradient approximators for discrete processes like Gumbel Softmax. (Yoon 2020) explains clearly that REINFORCE "directly encourages exploration of the policy towards the optimal solution". Our setting is similar where there is a mixup value network and an independent black-box regression model. For each episode, we initialize the regression model, train it, and obtain the validation loss, which is used to train the mixup value network once. The Mixup process between the mixup value network and a regression model is not a discrete and differentiable process. We do not see a way to approximate the gradient and thus use REINFORCE. Note that REINFORCE is also used in other systems like Neural Architecture Search (Zoph & Le 2017) for similar reasons. Next, we emphasize that the main contribution of this paper is solving data augmentation for regression by extending Mixup techniques. The fact that our framework cannot use state-of-the-art RL techniques does not seem to devalue this contribution. We will clarify the core story of our research and properly place our work among related RL algorithms in our revision.
>
> Zoph & Le 2017: Neural architecture search with reinforcement learning
>
> Comment 2: Linearity assumption for classification
>
> Thanks for raising this point. The reason linearity works in classification is that the label distance is not sensitive to the data distance. Even if two examples have a large data distance, their label distance is still 0 or 1. Hence, there is more benefit in mixing examples regardless of their labels than trying to be more selective in which examples to mix as in our setting. In the original Mixup paper (Zhang 2018), mixing all examples outperforms Empirical Risk Minimization on many classification datasets. Now in regression, the setting is completely different where two distant examples may have arbitrarily-different labels, so mixing with nearest neighbors becomes essential. As shown in Table 2 in our paper, (Zhang 2018)'s Original Mixup is never the best method. We will add these clarifications in our revision.
>
> Comment 3: Use of model for physical process
>
> Thanks for the great point. Indeed, for 3-d semiconductors, we know such models do exist, but are not used due to their inaccuracies on real products. Whether these models are accurate enough to improve our framework is an interesting future work. We emphasize that our framework already shows state-of-the-art regression performance without such models. We will clarify in our revision.
>
> Comment 4: Meta-learning
>
> We mentioned that our framework uses meta-learning because it learns from the output of the regression model. We will clarify in our revision.
>
> Comment 5: Determining kNN options
>
> We view the set of kNN options as a hyperparameter and generate them using a linear or exponential series. We will clarify in our revision.

---

> > ### Comment · Reviewer_XF4Y · 2021-12-07
> > **Response**
> >
> > On comment 1:
> > "The Mixup process between the mixup value network and a regression model is not a discrete and differentiable process"
> > What do you mean by this? The mixup network outputs probabilities that are used to sample examples to be mixed. Can I not approximate the gradients through this discrete process via Gumbel-softmax?
> >
> > My main issues here is we're not getting a close look at alternatives. REINFORCE might be an obvious choice, I agree, but there are other ways to encourage "exploration", for instance my adding stochasticity to the mixup value network. I feel like the choice to frame this whole thing as "RL" misplaces the model w.r.t. more relevant works, and this hurts the presentation.
> >
> > Comment 2:
> >  I understood this point from the paper. My concern was on the wording "arbitrarily-different labels". How are regression labels more potentially arbitrarily different than discrete? What does "arbitrary" mean?
> >
> > Comment 3:
> > It's fine if you move away from these physics examples. We know some of these models are inaccurate, sure, but are they worse than neural networks trained with REINFORCE?
> >
> > Comment 4:
> > I see that there's an inner-loop and outer-loop in training the regression model versus the mixup network. Was this important? (I don't recall seeing this in the paper)
> >
> > Comment 5:
> > OK

---

> > > ### Author Response · Authors · 2021-12-09
> > > **Response to Reviewer XF4Y**
> > >
> > > We appreciate your additional comments.
> > >
> > > Comment 1
> > >
> > > We clarify that using Gumbel-softmax is not enough to approximate the gradients.
> > >
> > > Here is the entire process described in four steps:
> > > 1. (x, y, k) -> mixup value network -> probability -> sampling -> mixup sampled pairs with the training dataset (mixed dataset)
> > > 2. Initialize regression model -> train regression model with mixed dataset and training dataset
> > > 3. x_valid, y_valid -> regression model -> validation loss
> > > 4. Update Mixup value network with the validation loss using REINFORCE
> > >
> > > We agree that the sampling process is discrete (sample or not sample), and the gradient can be propagated through sampling via Gumbel-softmax, but we can train the mixup value network without REINFORCE only when all the above steps from the validation loss to the sampling are differentiable. However, there are three places where the process is non-differentiable. In Step 1, mixing sampled pairs with the training dataset is probably non-differentiable because the selected examples are mixed with training examples that are not contained in the input batch. In Step 2, training the regression model is non-differentiable because we need to initialize the regression model, sample batches, get outputs, initialize the gradient to zero (zero_grad), and update the weights of the regression model for each epoch. In addition, the training involves non-differentiable processes such as sampling a batch or updating weights (replacing old weights with newer ones). Finally in Step 3, the validation loss calculation is non-differentiable because it is computed on a separate validation set instead of using the mixed dataset and the outputs of the mixup value network. That is, the calculation is independent of the other processes. Hence, we do not see a way to approximate the gradient, which is why we used REINFORCE.
> > >
> > > We will add the necessary explanations in the revision. Please note that Neural Architecture Search (Zoph & Le 2017) simply argues that its reward signal is non-differentiable when justifying the usage of REINFORCE.
> > >
> > > Comment 2
> > >
> > > By arbitrary, we meant that the difference between two labels can be any value. In the discrete case, labels are either the same or different. In regression, labels can have a difference of say 1, 10, or any larger value. We will clarify this point in the revision.
> > >
> > > Comment 3
> > >
> > > For our 3-d semiconductor application, using a simulator was not even considered by our industry collaborators due to its poor performance. We suspect that the modeling is extremely difficult due to the complex physics. For other applications, simulators could perform better, but there is no guarantee. We will clarify in the revision.
> > >
> > > Comment 4
> > >
> > > You are right. Training the mixup value network is the outer-loop, while training the regression model is the inner-loop. We train the mixup value network to minimize the validation loss calculated by the regression model. Hence, the mixup value network is learning from the regression model's output. For this reason, our framework is performing meta-learning. We will emphasize this point in the revision.

---

### Decision · Program_Chairs · 2022-01-20

**Decision:**

Reject

**Comment:**

This paper generalize the idea of Mixup-based data augmentation for regression. Compared to classification for which Mixup was used, the paper argues that in regression the linearity assumption only holds within specific data or label distances. The paper thus proposes MixRL to select suitable pairs using k-nearest neighbor in a batch for mixup. The selection policy is trained with meta-learning by minimizing the validation-set loss. The approach provides consistent but small improvement over mixup on several datasets. Reviewers have also suggested discussion and comparison with more baselines, such as respective method using other (lower-variant) gradient estimators (e.g., gumbel-softmax), and using local input/output kernels for data selection, etc.